# The Preventive Role of mRNA Vaccines in Reducing Death among Moderate Omicron-Infected Patients: A Follow-Up Study

**DOI:** 10.3390/v14122622

**Published:** 2022-11-24

**Authors:** Amy Ming-Fang Yen, Sam Li-Sheng Chen, Chen-Yang Hsu, Tony Hsiu-Hsi Chen

**Affiliations:** 1School of Oral Hygiene, College of Oral Medicine, Taipei Medical University, Taipei City 110, Taiwan; 2Daichung Hospital, Miaoli County 350, Taiwan; 3Institute of Epidemiology and Preventive Medicine, College of Public Health, National Taiwan University, Taipei City 110, Taiwan

**Keywords:** COVID-19, mRNA, Omicron, survival, preventive effect

## Abstract

Very few studies have been conducted to assess the potential preventive role of vaccines, particularly mRNA vaccines, in the improvement of survival among moderate and severe hospitalized patients with COVID-19. After community-acquired outbreaks of the Omicron variant from 18 March until 31 May 2022, occurred in Taiwan, this retrospective cohort of 4090 moderate and 1378 severe patients admitted to hospital was classified according to whether they were administered an mRNA-based vaccine, and followed up to ascertain rates of death in both the vaccinated (≥2 doses) and unvaccinated (no or 1 dose) groups. The age-adjusted hazard ratio (aHR) of less than 1 was used to assess the preventive role of mRNA vaccines in reducing deaths among moderate and severe Omicron-infected patients. Survival was statistically significantly better for the ≥2 dose jab group (aHR, 0.75, 95% confidence interval [CI], 0.60 to 0.94) and even higher among those who had received a booster jab (aHR, 0.71; 95% CI, 0.55 to 0.91) compared with the unvaccinated group among moderate patients, but not among severe patients. In conclusion, unveiling the role of mRNA vaccines in preventing moderate but not severe COVID-19 patients from death provides new insights into how mRNA vaccines play a role in the pathway leading to a severe outcome due to Omicron COVID-19.

## 1. Introduction

Coronavirus disease 2019 (COVID-19) patients infected with the Omicron (B.1.1.529 variant) variant of concern (VOC) were less likely to experience a fatal outcome than those infected with other Severe Acute Respiratory Syndrome Coronavirus 2 (SARS-CoV-2) variants, such as the Alpha and Delta VOCs [1]. In addition to the biological properties of Omicron, with reduced pathogenicity related to compromising cell entry in TMPRSS2-expressing alveoli cells and compromising syncytia formation [2], another major reason for this was widespread administration of vaccines, which was expected to result in more mild or asymptomatic cases, although vaccination may not fully protect a person from Omicron infection because of evasion of vaccine-elicited neutralizing antibodies [3]. Although the efficacy of the vaccines has been demonstrated in reducing hospitalization and deaths [1], one of main concerns is that there is still a fraction of patients infected with the Omicron variant that are vulnerable to becoming moderate or severe cases, leading to death even after the provision of medical care and treatment during hospitalization. While the efficacy of vaccines in terms of protection from VOC infections may be attenuated as a result of mutations in the spike protein, slowing disease progression may be able to modulate the inflammatory process through the interferon-*r* signaling pathway, which has been previously reported when using an mRNA-based vaccine for cancer prevention [4,5]. Very few studies have been conducted to assess the potential preventive role of vaccines related to survival among hospitalized patients. It was therefore interesting to examine the survival of the vaccinated group and the unvaccinated group in moderate and severe patients infected with the Omicron variant. We found that the vaccinated group, particularly those vaccinated with an mRNA vaccine, showed better survival rates than the unvaccinated group in moderate patients, but similar survival in severe patients, based on the empirical data from a series of community-acquired outbreaks of the Omicron VOC among a susceptible population administered mRNA vaccines in Taiwan.

## 2. Materials and Methods

### 2.1. Study Population

The study population consisted of a total of 4090 moderate and 1378 severe patients admitted to hospital after a series of community-acquired outbreaks of the Omicron variant from 18 March until 31 May 2022, in Taiwan. The retrospective cohort study was conducted to elucidate the relationship between the administration of vaccines and risk of death among moderate and severe patients upon admission to hospital.

### 2.2. Type of Vaccine for Primary and Booster Vaccination

As of 31 May 2022, the types of vaccine used in Taiwan included 63.7% (31.8 million doses) of the mRNA type (BNT-162b2 and mRNA-1273), followed by 30.7% (15.3 million doses) of the vector type (AZD-1222), and only 5.6% (2.8 million doses) of the protein subunit (MVC-COV1901) type. Following the temporal sequence of each type of vaccine available in Taiwan, the vector-based vaccine was mainly used for the primary series (2-dose) for health care workers and a small fraction of the elderly population aged 65 years and older, and mRNA-based vaccines were used both for primary series targeting the majority of the general population and for the booster dose (the third dose following the primary two-dose program), targeting the entire population [6]. Up to 31 May 2022, the coverage rates for the primary dose only and booster dose were 17.0% and 64.4%, respectively. The majority of mRNA vaccines used for primary and booster vaccination here provided an opportunity to assess whether mRNA vaccines play a significant preventive role in the improvement of survival.

### 2.3. Moderate and Severe Omicron-Infected COVID-19

Guided by the definition for case severity proposed by the World Health Organization [7] and National Institutes of Health [8], patients with any of SpO2 < 94% on room air, respiratory rate > 30 breaths/min, PaO2/FiO2 < 300, or lung infiltration > 50% on plain film were considered as moderate COVID-19 disease. A severe disease state of COVID-19 was defined for patients with any of the following presentations: lung infiltration > 50%, PaO2/FiO2 ≤ 300, requiring high flow oxygenation therapy provided by mask or non-invasive positive pressure ventilator, requiring mechanical ventilation or extracorporeal membrane oxygenation (ECMO), or clinical evidence of organ failure [9]. All of the COVID-19 patients with moderate and severe disease status were hospitalized to receive standard care following the national treatment guidelines [9]. 

The Taiwan Central Epidemic Command Center (CECC) reported that the resurgence of the Omicron variant in Taiwan started from 18 March 2022. The cumulative COVID-19 cases increased to 2,005,338 on 31 May 2022. Tabular data used for the following analysis were available from Taiwan CECC, including moderate and severe disease status on hospital admission, age, vaccination status, the date of discharge, and the date of death. Note that the authority of CECC was ensured by the Communicable Disease Control Act. Report of every confirmatory case to the CECC was mandatory. The Centers of Disease Control was responsible for maintaining this nationwide dataset, which is publicly available from the official website [10]. 

### 2.4. Statistical Analysis

To test whether the survival of Omicron-infected hospitalized patients in the two groups, defined by vaccination status, modified by the severity of disease following the postulate mentioned earlier, an interaction term between vaccination status and severity was tested by using a Cox proportional hazards regression model. Once such an effect modification was found to be statistically significant, the respective therapeutic effects of the mRNA vaccine on the survival of moderate and severe patients were plotted by using the Kaplan–Meier method with the log-rank test. The corresponding age-adjusted hazard ratios and 95% confidence interval (CI) were estimated by using the Cox proportional hazards regression model. A two-sided *p* value less than 0.05 was considered statistically significant. All analyses were conducted using SAS version 9.4 (SAS Institute, Cary, NC, USA).

## 3. Results

Table 1 shows the frequencies of hospitalized patients infected with the Omicron VOC by severity (moderate and severe), age, vaccination status, and fatality rates in Taiwan. Half of patients in both the moderate and severe series were more than 80 years old. The proportion who had received the primary two-dose and booster dose was 14% and 37% in the moderate patients, and 15% and 31% in the severe patients. The fatality rates among the moderate and severe patients were 7.4% and 79.9%. The fatality rate increased with age in both the moderate and severe patients. For moderate patients, the fatality rate was highest in the unvaccinated group (no or one dose), followed by the two-dose and booster vaccinated cases. However, in the severe patients, the fatality rate did not decrease with vaccine dose. 

We first tested whether the effect of vaccines on reducing death would be modified by the disease severity. Appendix A shows the estimated regression coefficients of the interaction term and the main effect of both vaccination and the severity of disease with adjustment for age. The interaction term was statistically significant for two-dose or booster jab modified by the severity, as mentioned in Section 2 (Wald test, χ^2^_(1)_ = 5.04; *p* = 0.0248), suggesting the preventive effect of mRNA vaccines associated with the improvement of survival was modified by the severity of disease. Results of the effect of vaccines with adjustment of age in the moderate and severe patients are separately reported in Table 2. Primary two-dose vaccination led to a reduction of hazard of death by 25% among the moderate patients (aHR, 0.75; 95% CI, 0.60 to 0.94), but this was not seen in the severe patients (aHR, 1.00; 95% CI, 0.89 to 1.12).

Figure 1A showed the vaccinated group in moderate patients had statistically significantly higher survival than the unvaccinated group (Log-Rank test, χ^2^_(1)_ = 4.8931; *p* = 0.0270), whereas there was identical survival between the two groups for severe patients (Log-Rank test, χ^2^_(1)_ = 0.0003; *p* = 0.9873) (Figure 1B). 

Recall that the entire population in Taiwan had been invited to have a booster vaccination with the mRNA vaccine. This provided an opportunity to assess the pure therapeutic effect of mRNA vaccination. When the booster group was separated from the primary two-dose group among the vaccinated group, the interaction was statistically significant only for the booster (Wald test, χ^2^_(1)_ = 7.4521; *p* = 0.0063), but not for the primary two-dose group (Wald test, χ^2^_(1)_ = 0.0436; *p* = 0.8345), suggesting that the preventive effect of mRNA associated with death was modified by the severity of disease. Table 2 also shows the age-adjusted hazard ratio for the booster group (administration of mRNA vaccine) as opposed to the unvaccinated group was 0.71 (95% CI, 0.55 to 0.91) for moderate patients, whereas the corresponding estimate was 1.05 (95% CI, 0.89 to 1.12) for severe patients. As far as primary two-dose vaccination (the mixture of adenovirus-vectored vaccines and mRNA vaccines) is concerned, the corresponding figures were 0.87 (95% CI, 0.63 to 1.22) for moderate patients and 0.90 (95% CI, 0.76 to 1.08) for severe patients. The estimated regression coefficients with consideration of the interaction term for the two-dose vaccination group and the severity of disease are presented in Appendix A. The separate survival curves for two-dose and booster vaccines among moderate and severe patients are presented in Appendix A.

## 4. Discussion

Using a Taiwanese cohort of moderate and severe hospitalized patients, we found the previously administered mRNA vaccine may have reduced the risk of vaccinated moderate cases dying by 25%. The booster group, using the mRNA jab, produced a more remarkable 31% reduction. However, such a preventive role was not seen in severe patients.

As a platform to trigger the immune response and replace functional proteins, mRNA-based vaccination has garnered attention since 1990 [11,12]. The clinical application of this novel platform as a preventive vaccine for cancer has revealed its potential for modulating the immune system through the signaling pathway of type I interferon-r, involved with CD4+ and CD8+ T cell lineage and Toll-like receptors [12,13]. Similar findings on cellular immunity have also been reported in the phase III clinical trials of mRNA-based vaccines against COVID-19 [13]. These findings, together with the cascade of pathogenesis in which the derangement of systematic inflammation centers INF-*r* and Toll-like receptors, provide biological plausibility for the preventive potential of mRNA-based vaccines against COVID-19 [5,14]. 

Despite some studies providing compelling evidence of the lower effectiveness of mRNA vaccines against Omicron infection than other variants [15,16,17], mRNA vaccines were highly associated with strong protection against hospital admissions due to the Omicron VOC [18,19]. As the loss neutralizing activity after two doses of SARS-CoV-2 vaccine for Omicron was more likely than for other variants such as Delta, boosting doses of mRNA vaccines reduced evasion of neutralization antibodies for protection against Omicron infection, which was a necessary vaccination policy [20,21]. In addition, previous studies reported the significant reduction of symptomatic COVID-19, severe illness, and COVID-19 death attributed to a third dose of mRNA COVID-19 vaccine against the Omicron variant [3,20]. Our finding regarding the improvement of survival following a booster jab further supports those previous findings on the effectiveness of booster vaccination.

It could be argued that the therapeutic role of the mRNA vaccine was confounded in this study by the use of antiviral therapy. By the end of May, only 4.0% (74,408 cases) of COVID-19 cases in Taiwan were prescribed oral antiviral agents, including nirmatrelvir/ritonavir and Molnupiravir. The empirical data used for the current analysis thus provided a unique opportunity to assess the sole preventive potential of mRNA-based vaccines with less contamination by anti-viral therapy, as they were collected before the widespread use of oral anti-viral therapies. 

There are some limitations in this study. Firstly, the use of only tabular data precluded us from considering individual characteristics, such as comorbidities. Because it is a nationwide universal program and the major disparity can mainly be attributed to age, we believe that the elderly people with a lower vaccination rate compared with younger people were also more likely to have underlying comorbidities. Moreover, testing the postulate by the classification of moderate and severe cases was intended to minimize the influence of co-morbidities on death, because severe cases are more likely to have underlying fatal co-morbid disease than moderate cases. Using 50 years or more of age as the proxy for having co-morbid chronic diseases, severe cases were more likely to be older than 50 compared to moderate cases, by 48% (odd ratio = 1.48 (1298 × 339)/(79 × 3751)). Comorbidities, more common in elderly people, have been reported to be associated with severe COVID-19 [22]. The adjustment for age and the classification of moderate and severe cases on this occasion may indirectly have adjusted for the influence of comorbidities. The preponderance of co-morbidity in severe cases may also account for why there was lack of a preventive role for vaccination among severe patients. However, further investigation, taking into account comorbidities, is still needed for confirmation. Secondly, we did not consider the time span between vaccine jabs and hospitalization. The influence from the waning of vaccines could not be explored. Third, our data were from the community outbreak when BA.2 was the dominant variant. However, the variant was not determined on an individual basis.

## 5. Conclusions

We reported on the preventive role of mRNA vaccines, associated with the reduction of death among moderate but not severe Omicron-infected patients. This requires further validation for different racial groups and regions. More importantly, this novel finding provides a new insight into how mRNA vaccines play a role in the pathway leading to the severity of and the risk of death from Omicron COVID-19.

## Figures and Tables

**Figure 1 viruses-14-02622-f001:**
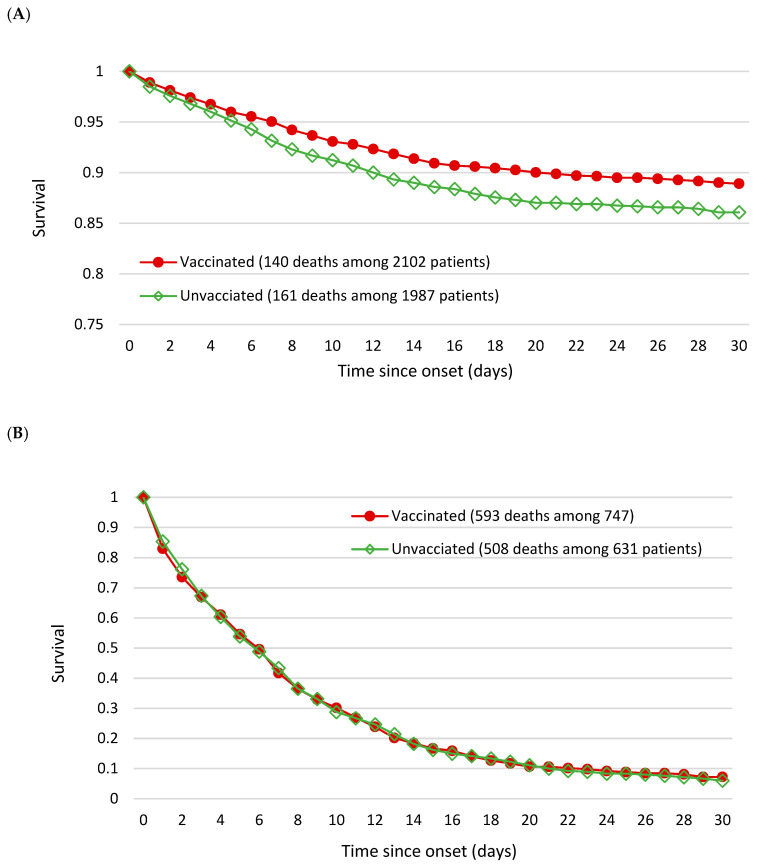
Survival curve of hospitalized patients by vaccine status (**A**) Moderate (**B**) Severe patients.

**Table 1 viruses-14-02622-t001:** Characteristics of hospitalized patients.

	Moderate				Severe			
	Patients	(%)	Death	Fatality (%)	Patients	(%)	Death	Fatality (%)
Total	4090	-	302	7.4	1378		1100	79.9
Age								
<10	61	1.5%	2	3.3	14	1.0%	6	42.9
10–29	74	1.8%	2	2.7	13	0.9%	7	53.8
30–49	204	5.0%	9	4.4	53	3.8%	35	67.3
50–69	844	20.6%	47	5.6	251	18.2%	185	73.7
70–79	911	22.3%	62	6.8	299	21.7%	246	82.3
80–89	1266	31.0%	103	8.1	434	31.5%	357	82.3
90+	730	17.8%	77	10.5	314	22.8%	264	84.1
Vaccination								
Unvaccinated (0 or 1 dose)	1988	48.6%	160	8.0	747	54.2%	592	79.3
Two-dose	590	14.4%	45	7.6	211	15.3%	160	75.8
Booster	1512	37.0%	95	6.3	420	30.5%	348	82.9

**Table 2 viruses-14-02622-t002:** Estimated regression coefficients with consideration of the interaction term of vaccination and the severity of disease with adjustment for age.

Variable	aHR	95% CI
Patients admitted in a moderate status		
Unvaccinated (0 or 1 dose)	(Reference)	
Vaccinated (2 or more doses)	0.75	0.60–0.94
Patients admitted in a severe status		
Unvaccinated (0 or 1 dose)	(Reference)	
Vaccinated (2 or more doses)	1.00	0.89–1.12
Patients admitted in a moderate status		
Unvaccinated (0 or 1 dose)	(Reference)	
Two-dose group	0.87	0.63–1.22
Booster group	0.71	0.55–0.91
Patients admitted in a severe status		
Unvaccinated (0 or 1 dose)	(Reference)	
Two-dose group	0.90	0.76–1.08
Booster group	1.05	0.92–1.20

## Data Availability

The datasets used in the current study and the source code are available from the authors on request.

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
