# Peer review of "The Preventive Role of mRNA Vaccines in Reducing Death among Moderate Omicron-Infected Patients: A Follow-Up Study"

_viruses, 2022, doi:10.3390/v14122622_

Round 1
Reviewer 1 Report (Previous Reviewer 2)
Table 1
Moderate severity / vaccination : 1987 +590 +1512 do not add up to 4090
Severe : 14 + 13 +52 + 251 + …… do not add up to 1378
Please define "unvaccinated"
Is having 1 dose equals unvaccinated ?
Abstract:
"In conclusion, unveiling the role of mRNA for preventing moderate COVID-19 patients from death strengthens scientific evidence on mRNA-based vaccine for cancer prevention."
I am afraid I do not see how this study can strengthen mRNA vaccine for cancer prevention
Conclusion:
More importantly, this novel finding provides a new insight into the mechanism of how mRNA vaccine plays a role in the pathway leading to the severity and the risk for death from Omicron-infected COVID-19
Prevention of COVID-related death by mRNA vaccine is known fact, proven repeatedly in many different studies, and I do not see how this study is related to vaccine mechanism or pathogenesis.
Author Response
Please see the attachment.

Reviewer 2 Report (Previous Reviewer 1)
All suggested revisions have been thoroughly corrected
Author Response
Thanks for your constructive comments during the revision of our manuscript.
Round 2
Reviewer 1 Report (Previous Reviewer 2)
no further comment
This manuscript is a resubmission of an earlier submission. The following is a list of the peer review reports and author responses from that submission.
Round 1
Reviewer 1 Report
In this study, Ming-Fan Yen et al. evaluated the effective role of vaccination, both after two doses and a booster dose, in preventing death among hospitalized patients with moderate or severe COVID-19 during a wave of SARS-CoV-2, Omicron variant, in Taiwan between March 18 and May 31, 2022.
The analysis showed that while vaccination was able to reduce the risk of death in subjects with moderate COVID-19 (aHR 0.75 for two-dose and 0.71 for booster jab), the same could not be postulated for cases of severe COVID-19. These data, supported by the high number of cases examined, highlights the efficacy of SARS-CoV-2 vaccine, in particular of the mRNA vaccine, in reducing the risk of death in subjects affected by COVID.
However, there are some major issues that should be addressed:
· An appropriate section in the text regarding the ethics statement should be added.
· I wonder if we can actually talk about a therapeutic effect of vaccination, rather than prevention of death. In fact, in order to discuss about a therapeutic effect, the vaccine should have been administered after the patients confirmation of SARS-CoV-2 positivity.
· It would be of relevant importance if data regarding the time elapsed since vaccination (two doses or booster dose) were available. In fact, it could be that COVID-19 severity may be due to a decrease in vaccine efficacy over time, which could explain the lack of difference in mortality rates between unvaccinated and vaccinated subjects. In that case, the distance from vaccination should correlate significantly with the rate of severity.
· As reported in the text, no data were available regarding the presence of comorbidities in the subjects examined. Unfortunately, I think this is a major shortcoming, especially for the evaluation of vaccine effectiveness among severe cases, as some of them may have been immunocompromised.
· In Figure 1a, it appears that the number of subjects analyzed for survival rate purposes among moderate patients (n. 4090) is only 140/2102 vaccinated (6.6%) and 161/1987 (8.1%) unvaccinated. I do not believe that the samples examined can be considered representative of the respective groups.
· Although it is reported that most subjects received mRNA vaccination, there are also many subjects (30.7%) who received vector-type vaccine as primary immunization. For this reason, it is difficult to generalize about the effectiveness of mRNA vaccination, especially among individuals who have received only two doses.
Minor issues:
Line 36: from Omicron infection
Line 41: from VOC infections
Lines 48-51: I would suggest reporting the concept not as a postulate, but as an indirect speech
Line 58-59: this sentence needs to be reformulated, as it is not clear
Lines 64-65: i would suggest to remove “the majority of” from the sentence
Line 94: Omicron should be written in upper case, as in the rest of the manuscript
Line 106: “fatality rates” would be more appropriate than “status of death”
Line 114: hospitalized patients
Line 156: the two-dose vaccinated group
Line 173: neutralizing activity
Figure 1: Among vaccinated subjects, two-dose and booster recipients should be reported separately.
Reviewer 2 Report
1/ The title misleading. By definition, therapeutic vaccine is the one administrated after a disease or infection has established. But this study is merely looking at the risk of death in the moderate and severe COVID-19 with/without vaccination. Similarly, "therapeutic role of vaccine" was repeatedly mentioned throughout the article, which I believe is referred to the protective effect of vaccination.
2/ The authors should clearly indicate the inclusion criteria, exclusion criteria, study period, the source of data, and the time frame used to define COVID-19 mortality. What is the coverage of CECC ? Is it a national database ? Is COVID-19 a reportable condition in Taiwan ? Also need to define "full-dose", "booster"
3/ The vaccination strategy in Taiwan 2.2 should be under the the session introduction
4/ 2.4 Two side p value less than 5% refers to a statistical significance ??
5/ Would the author comment on the lack of protective effective of booster dose on severe group ?
6/ Limitation of the study is not mentioned e.g. no individual vaccination data, variant determination, cause of death, past infection (hybrid immunity), etc ...
7/ "The elderly people with a lower vaccination rate compared with the younger people are also more likely to have the underlying comorbidities. Adjustment for age on this occasion may be sufficient"
I think the author should admit that not being able to take into account the comorbidities should be a major confounder. As in the severe group, its over-represented by elderly.